# Mathematical model of the dynamics of transmission and control of sporotrichosis in domestic cats

**Aurélio A. Araújo** [1]*, **Cláudia Codeço**[2], **Dayvison F. S. Freitas** [3], **Priscila M. de Macedo** [3], **Sandro A. Pereira**[3], **Isabella D. F. Gremião**[2], **Flávio Codeço Coelho**[1,4,5]

**1** Computational and Systems Biology, Fundação Oswaldo Cruz, Rio de Janeiro, Rio de Janeiro, Brazil, **2** Scientific Computing Program, Fundação Oswaldo Cruz, Rio de Janeiro, Rio de Janeiro, Brazil, **3** Instituto Nacional de Infectologia Evandro Chagas, Fundação Oswaldo Cruz, Rio de Janeiro, Rio de Janeiro, Brasil, **4** School of Applied Mathematics, Fundação Getulio Vargas, Rio de Janeiro, Rio de Janeiro, Brasil, **5** Institute of Global Health, University of Geneva, Geneva, Switzerland

* aurelio.aquino94@gmail.com

**Data Availability Statement:** All relevant data is in the article.

**Funding:** We acknowledge financial support from CAPES to AAA in the form of a PHD scholarship.

## Abstract

Sporotrichosis is a subcutaneous mycosis with a global distribution, also known as "rose gardener's disease". Brazil is experiencing a rapid spread of the zoonotic transmission of of *Sporothrix brasiliensis*, the main etiological agent of this disease in this country, affecting domestic felines. Cost-effective interventions need to be developed to control this emergent public health problem. To allow for the comparison of alternative control strategies, we propose in this paper, a mathematical model representing the transmission of *S. brasiliensis* among cats, stratified by age and sex. Analytical properties of the model are derived and simulations show possible strategies for reducing the endemic levels of the disease in the cat population, with a positive impact on human health. The scenarios included mass treatment of infected cats and mass implementation of contact reduction practices, such as neutering. The results indicate that mass treatment can reduce substantially the disease prevalence, and this effect is potentialized when combined with neutering or other contact-reduction interventions. On the other hand, contact-reduction methods alone are not sufficient to reduce prevalence.

## Introduction

Sporotrichosis is a neglected subcutaneous mycosis caused by species of *Sporothrix* Infections are typically caused by *S. brasiliensis*, *S. schenckii* or *S. globosa* [1, 2]. Although initially considered as a human disease, sporotrichosis was later described as a zoonotic disease [3, 4]. Countries with moderate to high burden include Brazil, Colombia, Peru and Mexico, in the Americas; South Africa, in Africa; and China, in Asia [5–7].

In Brazil, until the 1990's, most reports of human sporotrichosis were in adults working in activities like gardening and planting. There were also less frequent outbreaks linked to feline sporotrichosis, mainly involving cat owners and veterinarians [8]. Since then, feline

SAP is a CNPq Research Productivity Fellow (CNPq 312238/2020-7) and is supported by the State Funding Agency Fundação Carlos Chagas Filho de Amparo à Pesquisa do Estado do Rio de Janeiro (FAPERJ - E-26/201.737/2019). The funders had no role in study design, data collection and analysis, decision to publish, or preparation of the manuscript.

**Competing interests:** NO authors have competing interests Enter: The authors have declared that no competing interests exist.

sporotrichosis has emerged as an epidemic/endemic zoonotic disease of public health concern, initially in Rio de Janeiro, but nowadays, also in other states. According to Gremião and colleagues, from 1998 to 2015, approximately 5,000 human cases were reported by the national reference center for treatment of this disease [9]. The household is the main place of transmission, and cases are concentrated among mid-aged housewives, students and elderly individuals [10]. The disease is strongly under-reported both in animals and in humans.

Many animals can be infected by the fungus, including mice, rats, squirrels, cats and dogs [11], but infections in domestic and stray cats, have been increasingly described [7, 12–14]. Infected cats are the main source of infection for *Sporothrix sp.* in regions of zoonotic transmission. Behavioral aspects of the cat, such as fights involving scratching and biting, facilitate the transmission of this fungus [15, 16].

The strategies available to control the transmission of this fungus in the cat population and reduce risk of infection in humans, are of two types. One is focused in the screening and the treatment of infected individuals. The most common therapy is based on long-term administration of itraconazole as a monotherapy or in association with potassium iodide [17]. The second type is focused on the reduction of contact between susceptible and infected cats and the control of cat population size via neutering campaigns of females and males. Modeling the cost-effectiveness of these strategies is important to guide the development of a sporotrichosis control plan [18, 19].

In this paper, we present a mathematical description of the dynamics of sporotrichosis transmission in a population of domestic cats stratified by age and sex. The construction of the model is based on the literature on the ongoing feline sporotrichosis epidemics in Rio de Janeiro [12, 13, 17]. At the time of this writing, we found no published mathematical models of feline sporotrichosis transmission.

## The model

The model takes the form of a system of ordinary differential equations describing the population dynamics of a cat population in a large city (Fig 1). The cat population was stratified in two sexes (1 = male, 2 = female), and three age classes: kitten ($K$) as individuals less than one year of age, young ($Y$) as those with 1 to 5 years, and adult ($A$) as those with more years of age. The natural history of the disease was described by three states: susceptible ($S$), infected ($I$), and treated ($T$). Once infected, cats develop the disease and eventually die from the infection since there is no spontaneous cure. Recovery, however, can be attained when cats are treated, and once cured, they return to the susceptible class.

The demographic dynamics of the feline population is governed by a density dependent birth rate and density independent death rates. The birth rate, $b(t)$, is given by Eq 1 where $Y2 = S_{Y2} + I_{Y2} + T_{Y2}$ and $A2 = S_{A2} + I_{A2} + T_{A2}$.

$$b(t) = r\left(1 - \frac{N(t)}{\mathbb{C}}\right)(Y2 + A2).  \tag{1}$$

The parameter $\mathbb{C}$ represents the environment carrying capacity for the total domestic cat population, and $r$ is the per capita fertility rate, per month. Cats die by natural causes at age-dependent death rates $m_k$, $m_y$, $m_a$.

The transmission of *Sporothrix* between cats occurs by contacts motivated by fights between males or by sexual encounters between males and females or, less likely, by close contact between adults and kittens. We further assume that the young cats interact more frequently

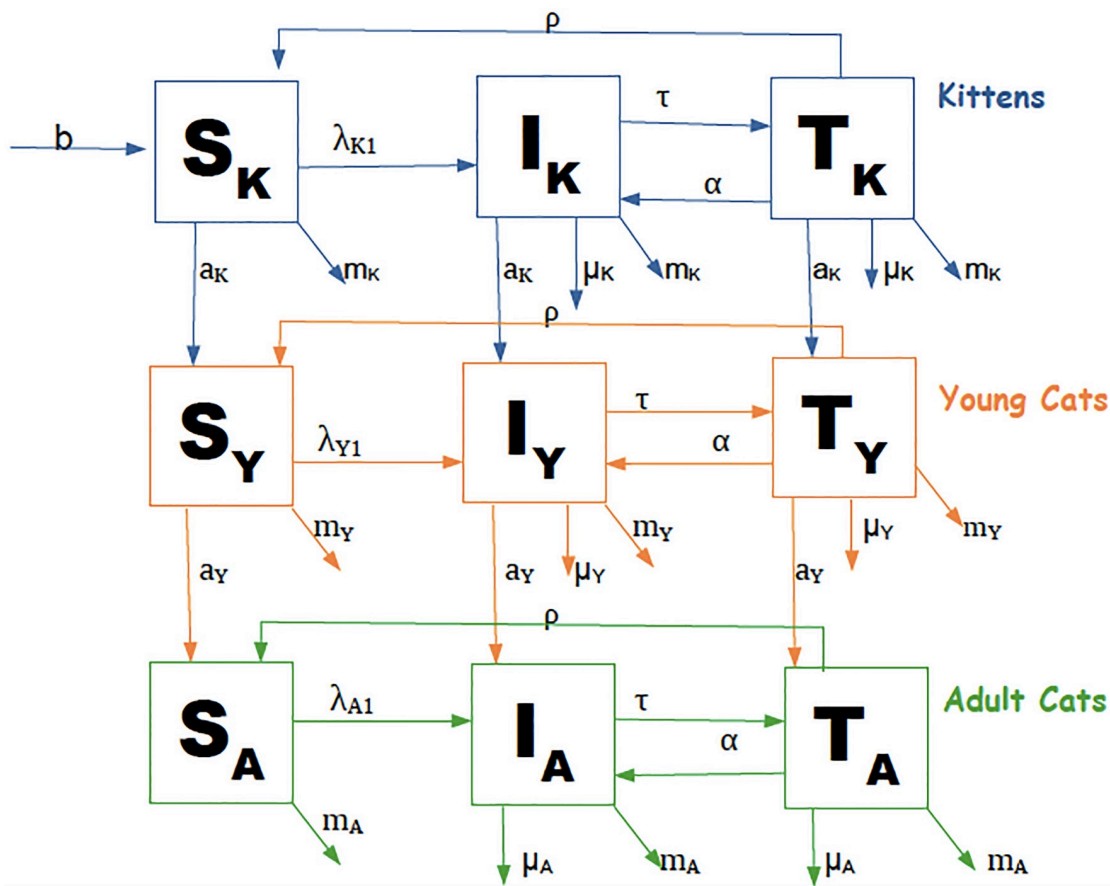

**Fig 1. Diagram of the transmission model of feline sporotrichosis.** Compartment structure is the same for both sexes.

than the older ones. To represent these different age and sex-specific modes of transmission, we defined three types of transmission rates $\beta_K$, $\beta_F$ and $\beta_X$. Transmission to kittens is governed by the expression $\beta_K = c_k p_k$ where $c_k$ is the rate of contact between kittens and other cats and $p_k$ is the probability of transmission through this contact. The transmission rate to young males through fighting is described by the expression $\beta_F = c_f p_f$, where $c_f$ is the fighting rate of the youngsters, and $p_f$ is the probability of transmission per fight. For adult males we multiply this transmission rate by $l$, an attenuation factor reflecting a lower fighting rate in adults. The transmission during sexual encounters is sex and age- specific. Transmission to young males and young females are described as $\beta_{X1} = c_{x1} p_{x1}$ and $\beta_{X2} = c_{x2} p_{x2}$, respectively, where $c_{xi}$ is the rate of encounters and $p_{xi}$ is the probability of transmission per contact. The constraints $c_{x1} > c_{x2}$ and $p_{x2} > p_{x1}$ imply that males have more sexual contacts than females, but the probability of acquiring infection is assumed to be higher in females. We further assume that older animals have less sexual contacts than the youngsters, which is represented by the attenuation factor $k1$ for males and $k2$ for females. The description of the parameters and the values used in the numerical analysis are given in Table 1.

Once the transmission rates are defined, the next step is to define the force of infection, which is the rate of new infections in each of the 6 sub-populations defined by the 3 age classes

**Table 1. Parameters of the model.** Values and definitions.

| Parameter | Values | Unit | Description |
|:---:|:---:|:---:|:---:|
| $\mathbb{C}$ | 325,918 | - | Environment carrying capacity [23] |
| $r$ | 10/12 | month$^{-1}$ | Average fertility rate [25] |
| $a_k$ | $\frac{1}{12}$ | month$^{-1}$ | Kitten aging rate |
| $a_y$ | $\frac{1}{36}$ | month$^{-1}$ | Young cat aging rate |
| $m_k$ | 0.010 | month$^{-1}$ | Kitten natural mortality rate |
| $m_y$ | 0.010 | month$^{-1}$ | Young cats natural mortality rate |
| $m_a$ | 0.030 | month$^{-1}$ | Adult natural mortality rate |
| $\rho$ | 0.16 | month$^{-1}$ | Cure rate through treatment |
| $\tau$ | varied | month$^{-1}$ | Treatment rate |
| $\mu_k$ | $\frac{1}{12}$ | month$^{-1}$ | Disease-induced mortality—Kittens |
| $\mu_y$ | $\frac{1}{12}$ | month$^{-1}$ | Disease-induced mortality—Young |
| $\mu_a$ | $\frac{1}{12}$ | month$^{-1}$ | Disease-induced mortality—Adults |
| $\alpha$ | 0.05 | month$^{-1}$ | Treatment abandonment rate |
| $c_k$ | 5 | month$^{-1}$ | Kitten contact rate |
| $p_k$ | 0.05 | - | Prob. of transmission upon contact (kittens) |
| $c_f$ | 2 | month$^{-1}$ | Fighting rate (young) |
| $l$ | 0.800 | - | Fighting attenuation factor (adults) |
| $p_f$ | 0.900 | - | Probability of transmission due to fighting |
| $c_{x1}$ | 5 | month$^{-1}$ | Rate of sexual contact (young males) |
| $k_1$ | 0.200 | - | Male sexual activity attenuation factor |
| $p_{x1}$ | 0.400 | - | Probability of transmission during sexual contact (males) |
| $c_{x2}$ | 5 | month$^{-1}$ | Sexual contact rate (young females) |
| $k_2$ | 0.400 | - | Female sexual activity attenuation factor (adults) |
| $p_{x2}$ | 0.800 | - | Transmission prob. during sexual contact (females) |

and 2 sex classes:

$$\lambda_{K1} = \lambda_{K2} = \frac{\beta_K I_*}{N} \tag{2a}$$

$$\lambda_{Y1} = \frac{\beta_K(I_{K1} + I_{K2}) + \beta_F(I_{Y1} + lI_{A1}) + \beta_{X1}(I_{Y2} + k_2 I_{A2})}{N} \tag{2b}$$

$$\lambda_{Y2} = \frac{\beta_F(I_{K1} + I_{K2}) + \beta_{X2}(I_{Y1} + k_1 I_{A1})}{N} \tag{2c}$$

$$\lambda_{A1} = \frac{\beta_K(I_{K1} + I_{K2}) + l\beta_F(I_{Y1} + lI_{A1}) + k_1\beta_{X1}(I_{Y2} + k_2 I_{A2})}{N} \tag{2d}$$

$$\lambda_{A2} = \frac{\beta_K(I_{K1} + I_{K2}) + k_2\beta_{X2}(I_{Y1} + k_1 I_{A1})}{N} \tag{2e}$$

where $I_* = I_{K1} + I_{K2} + I_{Y1} + I_{Y2} + I_{A1} + I_{A2}$.

The following equations describe the demographic and transmission dynamics in each age group:

## Kittens

Male and female kittens are born at a rate $b(t)$ in the susceptible class, $S_k$. Susceptible kittens can die of natural death or age into the young class. Infection during the first year of age occurs by contact with other animals at a rate $\beta_F$. Infected kittens, $I_k$ can die of natural infection or from sporotrichosis, or be treated and move to the $T_k$ compartment. Treated animals can die from natural causes, or abandon the treatment and become infected again. If the animal completes the whole treatment, it may recover from the infection and move back to the susceptible compartment, since the infection does not confer immunity.

$$\frac{dS_{K1}}{dt} = b(t)/2 - (m_K + a_K)S_{K1} + \rho T_{K1} - \lambda_{K1}S_{K1} \tag{3a}$$

$$\frac{dI_{K1}}{dt} = \lambda_{K1}S_{K1} + \alpha T_{K1} - (\tau + \mu_K)I_{K1} - (m_K + a_K)I_{K1} \tag{3b}$$

$$\frac{dT_{K1}}{dt} = \tau I_{K1} - (\rho + \alpha + \mu_K)T_{K1} - (m_K + a_K)T_{K1} \tag{3c}$$

$$\frac{dS_{K2}}{dt} = b(t)/2 - (m_K + a_K)S_{K2} + \rho T_{K2} - \lambda_{K2}S_{K2} \tag{3d}$$

$$\frac{dI_{K2}}{dt} = \lambda_{K2}S_{K2} + \alpha T_{K2} - (\tau + \mu_K)I_{K2} - (m_K + a_K)I_{K2} \tag{3e}$$

$$\frac{dT_{K2}}{dt} = \tau I_{K2} - (\rho + \alpha + \mu_K)T_{K2} - (m_K + a_K)T_{K2} \tag{3f}$$

## Young cats

Susceptible young males and females can die of natural cause, or age into the adult class. Young males can acquire infection from fight contacts with other youngsters or with adult males, the latter with lower probability. They can also acquire infection from sexual contacts with young and adult females, the former more frequently than the latter. As with the kittens, young infected individuals can die from other causes or from sporotrichosis, or be treated and move to the $T_Y$ classes. Treated animals can abandon treatment and return to the $I_Y$

compartment or recover and move to the $S_Y$ compartment.

$$\frac{dS_{Y1}}{dt} = a_K S_{K1} - (m_Y + a_Y)S_{Y1} + \rho T_{Y1} - \lambda_{Y1}S_{Y1} \tag{4a}$$

$$\frac{dI_{Y1}}{dt} = a_K I_{K1} - (m_Y + a_Y)I_{Y1} + \lambda_{Y1}S_{Y1} + \alpha T_{Y1} - (\tau + \mu_Y)I_{Y1} \tag{4b}$$

$$\frac{dT_{Y1}}{dt} = a_K T_{K1} - (m_Y + a_Y)T_{Y1} + \tau I_{Y1} - (\rho + \alpha + \mu_Y)T_{Y1} \tag{4c}$$

$$\frac{dS_{Y2}}{dt} = a_K S_{K2} - (m_Y + a_Y)S_{Y2} + \rho T_{Y2} - \lambda_{Y2}S_{Y2} \tag{4d}$$

$$\frac{dI_{Y2}}{dt} = a_K I_{K2} - (m_Y + a_Y)I_{Y2} + \lambda_{Y2}S_{Y2} + \alpha T_{Y2} - (\tau + \mu_Y)I_{Y2} \tag{4e}$$

$$\frac{dT_{Y2}}{dt} = a_K T_{K2} - (m_Y + a_Y)T_{Y2} + \tau I_{Y2} - (\rho + \alpha + \mu_Y)T_{Y2} \tag{4f}$$

## Adult cats

The equations describing the dynamics of adult cats are similar to the youngster's. Adults face the same modes of exposure but at lower rates, since adults are considered less active.

$$\frac{dS_{A1}}{dt} = a_Y S_{Y1} - m_A S_{A1} + \rho T_{A1} - \lambda_{A1}S_{A1} \tag{5a}$$

$$\frac{dI_{A1}}{dt} = a_Y I_{Y1} - m_A I_{A1} + \lambda_{A1}S_{A1} + \alpha T_{A1} - (\tau + \mu_A)I_{A1} \tag{5b}$$

$$\frac{dT_{A1}}{dt} = a_Y T_{Y1} - m_A T_{A1} + \tau I_{A1} - (\rho + \alpha + \mu_K)T_{A1} \tag{5c}$$

$$\frac{dS_{A2}}{dt} = a_Y S_{Y2} - m_A S_{A2} + \rho T_{A2} - \lambda_{A2}S_{A2} \tag{5d}$$

$$\frac{dI_{A2}}{dt} = a_Y I_{Y2} - m_A I_{A2} + \lambda_{A2}S_{A2} + \alpha T_{A2} - (\tau + \mu_A)I_{A2} \tag{5e}$$

$$\frac{dT_{A2}}{dt} = a_Y T_{Y2} - m_A T_{A2} + \tau I_{A2} - (\rho + \alpha + \mu_A)T_{A2} \tag{5f}$$

The full model presented above, in the absence of the disease can be reduced to the following 6 equations, describing the population dynamics of the cat population and its sex and age

structure.

$$\frac{dK_1}{dt} = 1/2(r(1 - N/C))(Y_2 + A_2) - m_k K_1 - a_k K_1 \tag{6a}$$

$$\frac{dY_1}{dt} = a_k K_1 - m_y Y_1 - a_y Y_1 \tag{6b}$$

$$\frac{dA_1}{dt} = a_y Y_1 - m_a A_1 \tag{6c}$$

$$\frac{dK_2}{dt} = 1/2(r(1 - N/C))(Y_2 + A_2) - m_k K_2 - a_k K_2 \tag{6d}$$

$$\frac{dY_2}{dt} = a_k K_2 - m_y Y_2 - a_k Y_2 \tag{6e}$$

$$\frac{dA_2}{dt} = a_y Y_2 - m_a A_2 \tag{6f}$$

**Model parameters.**   There is much uncertainty regarding many aspects of the demography of urban cats in areas where sporotrichosis is emerging. To choose reasonable values, we searched the literature and consulted with veterinarians with experience with this disease [20–22]. For the numerical analysis, we considered the cat population of the Rio de Janeiro city, Brazil, a city with 6.3 million people and 2, 177, 297 households. Assuming an estimate of one cat for every 19.33 humans [23], we estimate a total population of $N = 325, 918$ cats. This number may be underestimated since their study only accounted for domiciled animals. Female cats have 3-5 kittens per litter, with an average of 2-3 litters per year [24–26]. For this work, we considered 2 litters per year with 5 kittens each. The remaining parameters such as treatment rate ($\tau$), treatment abandonment rate ($\alpha$), cure rate ($\rho$), and death rate from the disease ($\mu$) were attributed values considered reasonable for the scenarios explored, as no reliable values were available for them.

**Control scenarios.**   We used the model to investigate two forms of control, mass pharmacological treatment and mass neutering. Pharmacological treatment is explicitly represented in the model by the parameters $\tau$ (treatment rate), $\alpha$ (treatment abandonment rate) and $\rho$ (cure rate). Neutering, on the other hand, is considered to affect mainly the contact rates for fighting and sex, $c_f$ and $c_{x[1,2]}$, respectively.

We considered scenarios where strategies were implemented to control the dissemination of the disease. Twelve scenarios were created, combining increased rates of treatment of infected individuals and reduced rate of contact between animals via methods such as neutering and avoiding cat's access to the outdoor environment.

## Results

### Demographic dynamics of the urban feline population

The demographic dynamics described in model (7) is shown in Fig 2 using values from Table 1. The age structure of the cat population at this equilibrium is 9.52% kittens, 21.01% young, and 19.45% adults for each sex. Sex ratio is assumed 1:1 at birth and there are no sex differences in natural death rates.

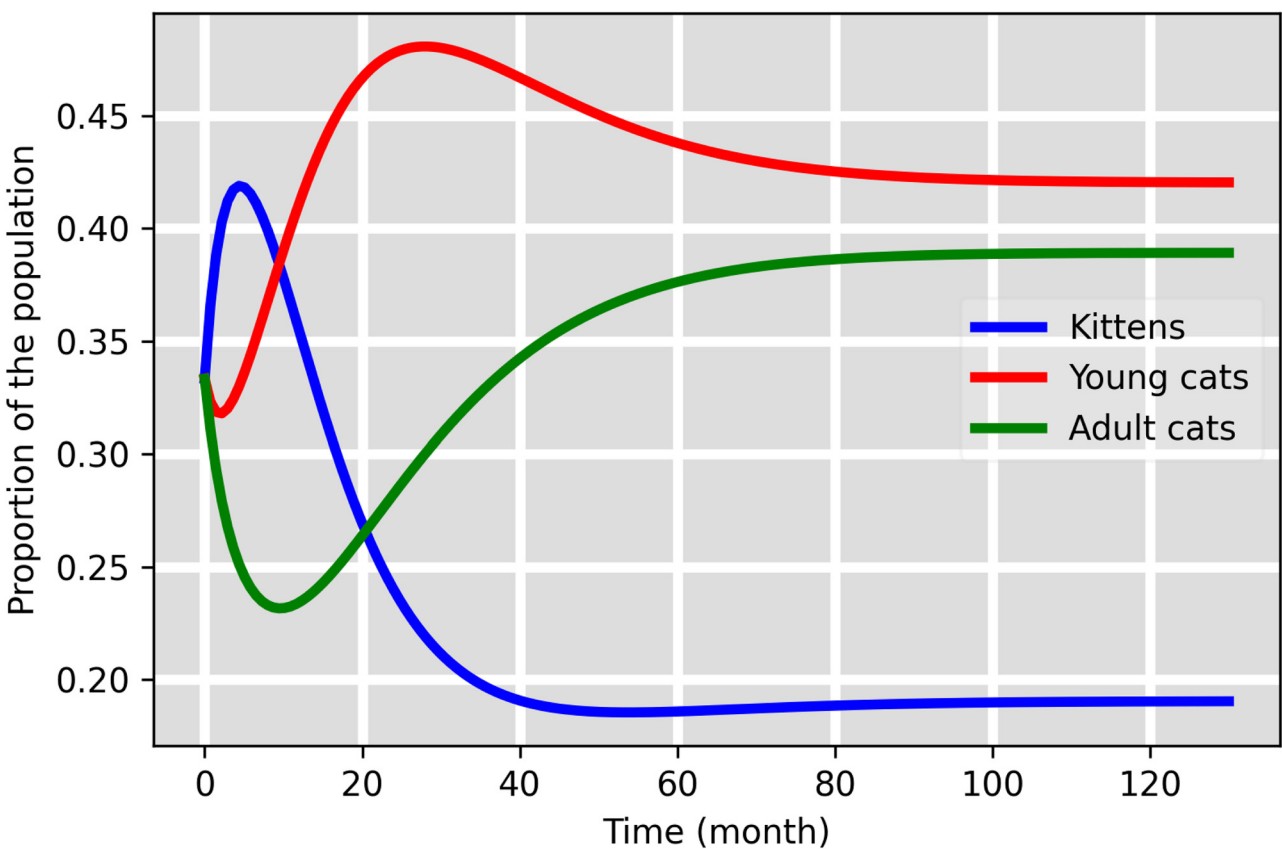

**Fig 2. Age distribution of the feline population in the equilibrium.**

We derived closed form expressions for the age-structure of the feline population at steady-state (Eq 7):

$$K_{1,2}^* = \frac{A_{1,2}^* m_a + Y_{1,2}^* m_y}{a_k}, \tag{7a}$$

$$Y_{1,2}^* = \frac{a_k m_a}{\delta}, \tag{7b}$$

$$A_{1,2}^* = \frac{a_k a_y}{\delta}, \tag{7c}$$

where $\delta = \frac{N}{2(a_k a_y + a_k m_a + a_y m_a + m_a m_y)}$.

This equilibrium (given by 7) is stable, as the dominant eigenvalue of the system's Jacobian matrix at this equilibrium is real and negative.

When simulating the introduction of sporotrichosis in this population, the cat population is assumed to be at this steady-state age-structure, being all individuals susceptible.

## The next-generation matrix

The basic reproduction number, $\mathcal{R}_0$, measures the rate of increase of incidence upon the arrival of an infected individual in a completely susceptible population. Values above one

indicate disease spread, while values below one indicate disease extinction. Deriving an expression for $\mathcal{R}_0$ is important since we can assess which parameters contribute the most to transmission. One method to derive the $\mathcal{R}_0$ is by computing the spectral radius of the next-generation matrix of the model. Following the methodology proposed by Van de Driessche [27], we have calculated the next-generation matrix, $M = FV^{-1}$, where the matrices $F$ and $V$ represent the flows at the disease-free equilibrium (Eq 7), assuming the population age distribution are at the steady-state given by Eq 6.

$$
F = \begin{bmatrix}
\dfrac{N\beta_k m_a(a_y + m_y)}{\delta} & \dfrac{N\beta_k m_a(a_y + m_y)}{\delta} & \dfrac{N\beta_k m_a(a_y + m_y)}{\delta} & \dfrac{N\beta_k m_a(a_y + m_y)}{\delta} & \dfrac{N\beta_k m_a(a_y + m_y)}{\delta} & \dfrac{N\beta_k m_a(a_y + m_y)}{\delta} \\[2mm]
\dfrac{Na_k\beta_k m_a}{\delta} & \dfrac{Na_k\beta_b m_a}{\delta} & \dfrac{Na_k\beta_b l m_a}{\delta} & \dfrac{Na_k\beta_k m_a}{\delta} & \dfrac{Na_k\beta_{x1} m_a}{\delta} & \dfrac{Na_k\beta_{x1} k_2 m_a}{\delta} \\[2mm]
\dfrac{Na_k a_y\beta_k}{\delta} & \dfrac{Na_k a_y\beta_b l}{\delta} & \dfrac{Na_k a_y\beta_b l^2}{\delta} & \dfrac{Na_k a_y\beta_k}{\delta} & \dfrac{Na_k a_y\beta_{x1} k_1}{\delta} & \dfrac{Na_k a_y\beta_{x1} k_1 k_2}{\delta} \\[2mm]
\dfrac{N\beta_k m_a(a_y + m_y)}{\delta} & \dfrac{N\beta_k m_a(a_y + m_y)}{\delta} & \dfrac{N\beta_k m_a(a_y + m_y)}{\delta} & \dfrac{N\beta_k m_a(a_y + m_y)}{\delta} & \dfrac{N\beta_k m_a(a_y + m_y)}{\delta} & \dfrac{N\beta_k m_a(a_y + m_y)}{\delta} \\[2mm]
\dfrac{Na_k\beta_k m_a}{\delta} & \dfrac{Na_k\beta_{x2} m_a}{\delta} & \dfrac{Na_k\beta_{x2} k_1 m_a}{\delta} & \dfrac{Na_k\beta_k m_a}{\delta} & 0 & 0 \\[2mm]
\dfrac{Na_k a_y\beta_k}{\delta} & \dfrac{Na_k a_y\beta_{x2} k_2}{\delta} & \dfrac{Na_k a_y\beta_{x2} k_1 k_2}{\delta} & \dfrac{Na_k a_y\beta_k}{\delta} & 0 & 0
\end{bmatrix}, \quad (8)
$$

$$
V = \begin{bmatrix}
\mathbb{O}_k & 0 & 0 & 0 & 0 & 0 \\
-a_k & \mathbb{O}_y & 0 & 0 & 0 & 0 \\
0 & -a_y & \mathbb{O}_a & 0 & 0 & 0 \\
0 & 0 & 0 & \mathbb{O}_k & 0 & 0 \\
0 & 0 & 0 & -a_k & \mathbb{O}_y & 0 \\
0 & 0 & 0 & 0 & -a_y & \mathbb{O}_a
\end{bmatrix}, \quad (9)
$$

where $\mathbb{O}_k = a_k + m_k + \mu_k + \tau$, $\mathbb{O}_y = a_y + m_y + \mu_y + \tau$, and $\mathbb{O}_a = m_a + \mu_a + \tau$, being the output flow from infected compartments at each age class.

The full matrix $M$ was omitted due to its size. A simplified version is presented instead, obtained by turning-off the treatment dynamics ($\tau = 0$ and $\alpha = 0$) and setting the aging rates to zero. The justification is that aging is a slower process compared to disease invasion and for the purposes of studying the instant of disease invasion, it could be ignored. The simplified

matrix $M_r$(Eq 10) is:

$$
M_r = \begin{bmatrix}
\dfrac{K^*\beta_k}{m_k+\mu_k} & \dfrac{K^*\beta_k}{m_y+\mu_y} & \dfrac{K^*\beta_k}{m_a+\mu_a} & \dfrac{K^*\beta_k}{m_k+\mu_k} & \dfrac{K^*\beta_k}{m_y+\mu_y} & \dfrac{K^*\beta_k}{m_a+\mu_a} \\[2ex]
\dfrac{Y^*\beta_k}{m_k+\mu_k} & \dfrac{Y^*\beta_b}{m_y+\mu_y} & \dfrac{Y^*\beta_b l}{m_a+\mu_a} & \dfrac{Y^*\beta_k}{m_k+\mu_k} & \dfrac{Y^*\beta_{x1}}{m_y+\mu_y} & \dfrac{Y^*\beta_{x1}k_2}{m_a+\mu_a} \\[2ex]
\dfrac{A^*\beta_k}{m_k+\mu_k} & \dfrac{A^*\beta_b l}{m_y+\mu_y} & \dfrac{A^*\beta_b l^2}{m_a+\mu_a} & \dfrac{A^*\beta_k}{m_k+\mu_k} & \dfrac{A^*\beta_{x1}k_1}{m_y+\mu_y} & \dfrac{A^*\beta_{x1}k_1k_2}{m_a+\mu_a} \\[2ex]
\dfrac{K^*\beta_k}{m_k+\mu_k} & \dfrac{K^*\beta_k}{m_y+\mu_y} & \dfrac{K^*\beta_k}{m_a+\mu_a} & \dfrac{K^*\beta_k}{m_k+\mu_k} & \dfrac{K^*\beta_k}{m_y+\mu_y} & \dfrac{K^*\beta_k}{m_a+\mu_a} \\[2ex]
\dfrac{Y^*\beta_k}{m_k+\mu_k} & \dfrac{Y^*\beta_{x2}}{m_y+\mu_y} & \dfrac{Y^*\beta_{x2}k_1}{m_a+\mu_a} & \dfrac{Y^*\beta_k}{m_k+\mu_k} & 0 & 0 \\[2ex]
\dfrac{A^*\beta_k}{m_k+\mu_k} & \dfrac{A^*\beta_{x2}k_2}{m_y+\mu_y} & \dfrac{A^*\beta_{x2}k_1k_2}{m_a+\mu_a} & \dfrac{A^*\beta_k}{m_k+\mu_k} & 0 & 0
\end{bmatrix}
\tag{10}
$$

Due to the structure of $M$, it was not possible to derive a closed form expression for $\mathcal{R}_0$, its spectral radius. But it is possible to calculate specific $\mathcal{R}_0$ numerical values, by substituting parameter values into $M$. For example, with the values from Table 1, $\mathcal{R}_0 = 1.227$.

The investigation of the matrix $M$ itself is quite useful. Elements $(i, j)$, $i, j \in \{Ik_1, Iy_1, Ia_1, Ik_2, Iy_2, Ia_2\}$ of $M$ can be interpreted as partial reproduction numbers of the compartments $i$ in response to the introduction of an infected individual in the compartment $j$. As such, one can compute $\Sigma_j M[i, j]$ as the partial basic reproduction number of compartment $i$, that is, the expected number of secondary cases in compartment $i$, after the introduction of one infected individual in any of the sex and age classes. Meanwhile, a sum across the rows of $M$, $\Sigma_i M[i, j]$, produces the expected number of secondary cases on the whole population after the introduction of one index case in compartment $j$.

## Sporotrichosis invasion in the absence of treatment

Fig 3 shows the epidemic curve of sporotrichosis after the arrival of one infected cat in the population with steady-state age structure, and $\mathcal{R}_0 = 1.227$. No treatment or other intervention is implemented. As the disease-free equilibrium is unstable, the infected population increases towards the endemic prevalence equilibrium, of approximately 70%, reached in a few years. The epidemic curve shape is typical of SIS models, that is, models of diseases without cure or immunity acquisition. Since all infected animals remain so until death, the observed prevalence results from the balance between infection and loss to natural and disease-induced mortality. In the absence of treatment, disease prevalence is greater among young cats, and among them, in males. This results from the higher exposure of this group to different modes of transmission.

In the absence of treatment, Fig 3 shows that the endemic equilibrium with 70% of prevalence can be reached within a few years. The mechanisms of transmission represented in the model, namely the role of fighting and sex in the transmission of sporotrichosis, lead to a greater importance of the young age class in establishing the *Sporothrix* endemic equilibrium.

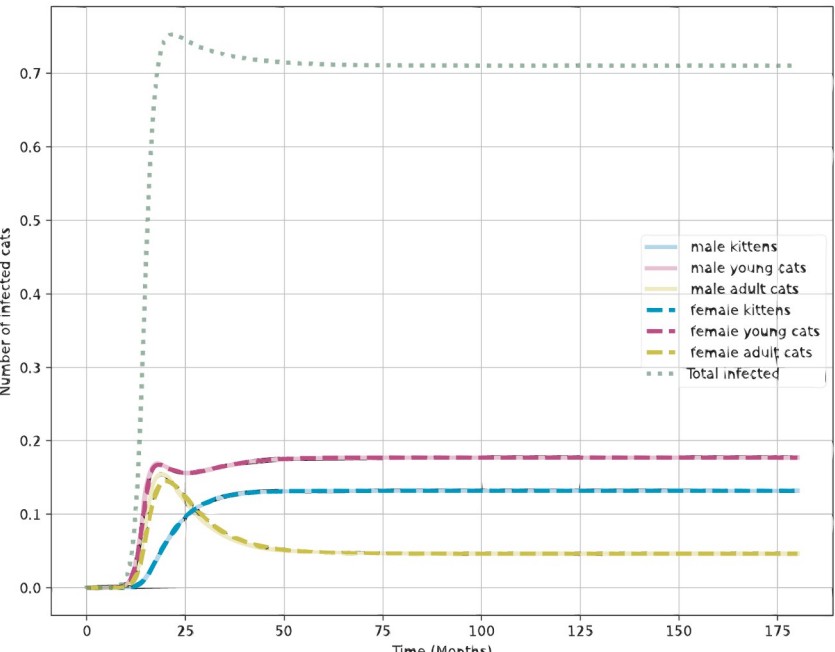

**Fig 3. Establishment of sporotrichosis on an entirely susceptible feline population at the demographic equilibrium shown in Fig 2, in the absence of controls.** The y-axis is scale to show the fraction of the entire population.

## Simulating control strategies

Table 2 shows the effect of control strategies on the prevalence of infection in 5 and 10 years. Important conclusions can be derived from this exercise. First, without treatment, prevalence will increase to the same equilibrium despite the implementation of contact reduction practices. Such practices affect the time to reach this equilibrium but not its magnitude. Treatment of 50% of the infected animals can reduce the prevalence to 11% (Fig 4) while 70% treatment

**Table 2. Approximate prevalence after 5 and 10 years, with different combinations of treatment and neutering.** NT: no treatment, NN: no neutering or other contact reduction measures, C50: 50% neutering. Treatment levels are specified by parameters $\tau$. *effect on sexual contacts, †effect on fighting.

| Scenario | Parameters | | | Outcomes | |
|---|---|---|---|---|---|
| - | $\beta_{x[1, 2]}$ | $\beta_f$ | $\tau$ | *Prevalence 5yrs* | *Prevalence 10 yrs* |
| NT, NN | $\beta_{x[1, 2]}$ | $\beta_f$ | 0 | 71% | 71% |
| NN | $\beta_{x[1, 2]}$ | $\beta_f$ | 0.5 | 11% | 11.5% |
| NN | $\beta_{x[1, 2]}$ | $\beta_f$ | 0.7 | 5% | 5.8% |
| C50* | $0.5\beta_{x[1, 2]}$ | $\beta_f$ | 0 | 70% | 70% |
| C50* | $0.5\beta_{x[1, 2]}$ | $\beta_f$ | 0.5 | 1% | 5.5% |
| C50* | $0.5\beta_{x[1, 2]}$ | $\beta_f$ | 0.7 | 0% | 0% |
| C50† | $\beta_{x[1, 2]}$ | $0.5\beta_f$ | 0 | 71% | 71% |
| C50† | $\beta_{x[1, 2]}$ | $0.5\beta_f$ | 0.5 | 8% | 9% |
| C50† | $\beta_{x[1, 2]}$ | $0.5\beta_f$ | 0.7 | 0% | 0.5% |
| C50*, † | $0.5\beta_{x[1, 2]}$ | $0.5\beta_f$ | 0 | 70% | 70% |
| C50*, † | $0.5\beta_{x[1, 2]}$ | $0.5\beta_f$ | 0.5 | 0% | 0% |
| C50*, † | $0.5\beta_{x[1, 2]}$ | $0.5\beta_f$ | 0.7 | 0% | 0% |

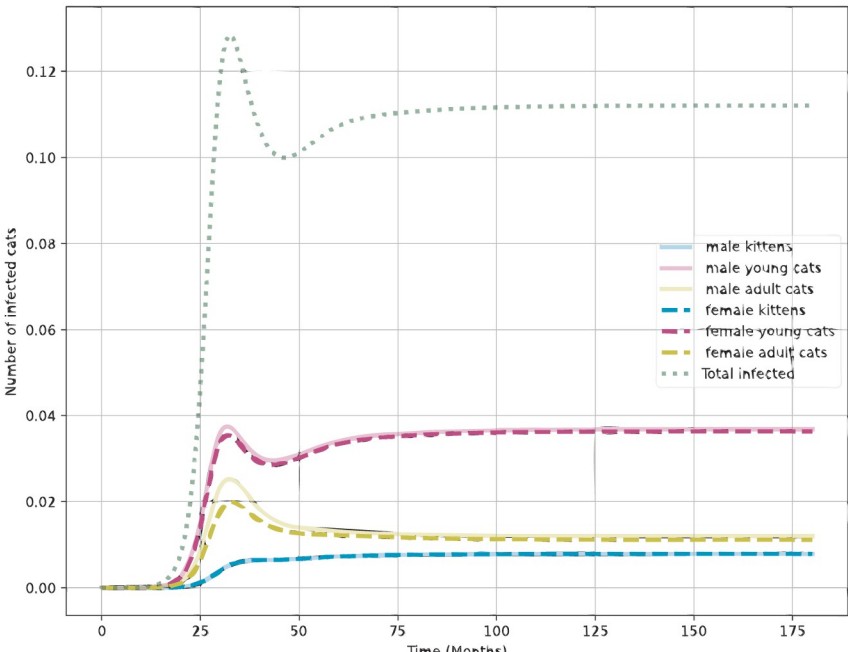

**Fig 4. Effect of a treatment rate of 0.5 on the endemic prevalences.** Y-axis represents fraction of the entire population.

coverage can reduce to 5% in 5 years without the need for further interventions. Disease elimination can be achieved by combining 50% of contact reduction and 50% of treatment.

## Discussion

Sporotrichosis is a neglected disease in Brazil, and public health policies and strategic plans are lacking. Developing cost-effective strategies to deal with this threat is paramount as the epidemics establishes in more cities every year. This is the first model of this disease, according to our knowledge. The model proposed seeks to represent the mechanisms of transmission described in the literature [13] and by clinical observations by the authors in their clinical practice in Rio de Janeiro, Brazil. The resulting dynamics provides a very important insight into the dynamics of sporotrichosis, and could be used to inform the design of future field studies to provide more accurate measures of some parameters of the model.

Although it was not possible to derive a closed form expression for the $\mathcal{R}_0$ of sporotrichosis, numeric values of $\mathcal{R}_0$ can be obtained by substituting the parameter values in the next-generation matrix. Our analysis shows that sporotrichosis can establish itself very quickly in the urban population of cats with a large proportion of non-neutered animals. In such a scenario, having pharmacological treatment available can substantially slow down the establishment of the disease in a population.

Pharmacological treatment of cats with sporotrichosis, however, faces many challenges. It is expensive and requires a long period (median = 4 months) of daily administration of oral antifungal drugs. This could be difficult to implement effectively at large scale and specially in low-income regions [18]. We show that simpler interventions such as mass neutering of male and female cats can synergistically increase the effectiveness of treatment as a public health strategy. This possibility opens an avenue for the development of more integrated approaches to disease control in the cat population.

The analyses presented in this paper are limited by the uncertainty of the parameter values. More field and laboratory studies are needed to elicit more realistic values for the natural history of the disease, as well as the population dynamics of urban cats. Nevertheless, the model is already useful to explore comparative scenarios, such as control strategies combining neutering and pharmacological treatment, versus each control separately. Our results support the expected conclusion that the most effective way to keep the prevalence of sporotrichosis in any cat population is to have a quick response both in terms of diagnosis and treatment of the cats as soon as they show the first symptoms of the disease. The availability of free neutering services can aid to drive the disease to undetectable levels.

## Author Contributions

**Conceptualization:** Aurélio A. Araújo, Cláudia Codeço, Dayvison F. S. Freitas, Priscila M. de Macedo, Sandro A. Pereira, Isabella D. F. Gremião, Flávio Codeço Coelho.

**Formal analysis:** Aurélio A. Araújo, Cláudia Codeço, Flávio Codeço Coelho.

**Funding acquisition:** Dayvison F. S. Freitas, Priscila M. de Macedo, Sandro A. Pereira, Isabella D. F. Gremião.

**Methodology:** Aurélio A. Araújo, Cláudia Codeço, Flávio Codeço Coelho.

**Writing – original draft:** Aurélio A. Araújo, Cláudia Codeço, Flávio Codeço Coelho.

**Writing – review & editing:** Aurélio A. Araújo, Cláudia Codeço, Dayvison F. S. Freitas, Priscila M. de Macedo, Sandro A. Pereira, Isabella D. F. Gremião, Flávio Codeço Coelho.

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
