## [Decision Letter · Decision Letter 0]

12 Sep 2022

PONE-D-22-19278

Transmission dynamics and control of Sporotrichosis in domestic felines

PLOS ONE

Dear Dr. Araújo,

Thank you for submitting your manuscript to PLOS ONE. After careful consideration, we feel that it has merit but does not fully meet PLOS ONE’s publication criteria as it currently stands. Therefore, we invite you to submit a revised version of the manuscript that addresses the points raised during the review process.

As you can see in the reports, both reviewers point to issues in the model assumption, the validity of the approach, and the connection with real scenarios and data. Address all reviewers' concerns before submitting a revised version.

We look forward to receiving your revised manuscript.

Kind regards,

Sebastián Gonçalves, Ph.D.

Academic Editor

PLOS ONE

Journal Requirements:

2. Please note that PLOS ONE has specific guidelines on code sharing for submissions in which author-generated code underpins the findings in the manuscript. In these cases, all author-generated code must be made available without restrictions upon publication of the work. Please review our guidelines at https://journals.plos.org/plosone/s/materials-and-software-sharing#loc-sharing-code and ensure that your code is shared in a way that follows best practice and facilitates reproducibility and reuse. New software must comply with the Open Source Definition.

 "The author(s) received no specific funding for this work.

We acknowledge financial support from CAPES to AAA in the form of a PHD scholarship. SAP is a CNPq Research Productivity Fellow (CNPq 312238/2020-7) and is supported by the State Funding Agency Fundação Carlos Chagas Filho de Amparo à Pesquisa do Estado do Rio de Janeiro (FAPERJ - E-26/201.737/2019)." 

6. Please ensure that you include a title page within your main document. You should list all authors and all affiliations as per our author instructions and clearly indicate the corresponding author.

7. Please remove your figures from within your manuscript file, leaving only the individual TIFF/EPS image files, uploaded separately.  These will be automatically included in the reviewers’ PDF.

Reviewers' comments:

Reviewer's Responses to Questions

**Comments to the Author**

1. Is the manuscript technically sound, and do the data support the conclusions?

Reviewer #1: Partly

Reviewer #2: Partly

2. Has the statistical analysis been performed appropriately and rigorously? 

Reviewer #1: Yes

Reviewer #2: Yes

3. Have the authors made all data underlying the findings in their manuscript fully available?

Reviewer #1: Yes

Reviewer #2: Yes

4. Is the manuscript presented in an intelligible fashion and written in standard English?

Reviewer #1: Yes

Reviewer #2: Yes

5. Review Comments to the Author

Reviewer #1: The paper entitled "Transmission dynamics and control of Sporotrichosis in domestic felines"

deals with the dynamics of the Sporotrichosis mycosis on felines. The approach is based on

a Susceptible, Infected and Treated compartment model, divided in three age categories and sex.

Force of infection is based on the feline behavior according to age. Interventions are considered

in order to mitigate the evolution of the disease such as treatments of felines with the disease,

reduction of contacts between susceptible and infected cats and neutering.

The model parameters are obtained from empirical data from the city of Rio de Janeiro in Brazil

and additional suppositions.

Approach is certainly interesting and original for this diseases, but there are some points

that need to be considered before publication:

1) In page 4 the authors state that the probability of acquiring the infection is higher in females. Why?

2) In section 3 the distribution of disease prevalence is obtained for the different groups considered

in the model. Are there any empirical data on this for some comparison?

3) In the pdf, just after the figure for the flow chart of the model, there is a figure without any caption,

which is repeated a few pages later. Please provide a caption and refer it in the text properly.

4) For me the main shortcoming of the paper is the absence of a discussion of how the control of disease is

implemented in the model. Is it turned on at some moment in time and set at some constant rate? Is it

implemented just once? How do they change the model equations? Section 3.4 should be expanded with some discussion

and details on this.

Otherwise, I believe the paper can be published if the above points are addressed.

Reviewer #2: Review

PONE-D-22-19278

Transmission dynamics and control of Sporotrichosis in domestic felines

The authors formulate a mathematical model to simulate the population dynamics of a zoonotic disease transmission among domestic cats in Rio de Janeiro city, Brazil. They build systems of differential equations for three age classes of cats (kittens, young and adults) for males and females, as well as for the whole population. With the system of equations they simulate the dynamics of disease transmission among domicile cats and they obtain the variability of the number of infected cats in time. Finally they assess the disease prevalence in cats, after the implementation of control treatments in 12 scenarios.

This is a novel study, the methods used are standard and well implemented. I recommend the publication of the manuscript after the authors address the following concerns:

Major comments:

1) One of my main concern is regarding the mean field formulation of the model, because as the authors point out they took into account only domiciled cats (as opposed to stray cats), and therefore those cats might be highly clustered inside houses with very low interaction among them? Therefore the spatial distribution of cats might be highly relevant. Can the authors justify a bit more why the mean field approach is acceptable?

2) Related with the above concern. It would be of importance to describe at some point in the manuscript, the behavior and area of movement of these domiciled cats. Can they circulate outside their houses through the neighborhood? If so, they might encounter other domestic or stray cats on the way. Those stray cats are not considered into the model...

Otherwise, where do the encounters occur among domiciled cats? In the same house? How many cats are on average in houses? I think that it’s necessary to include in the manuscript a brief answer to these questions because they are relevant to disease transmission.

3) page 4, in the model the probability of acquiring infection is assumed to be higher in females. Is there any reference the authors may cite to support this assumption? The same for the assumption that males have more sexual encounters than females. Please provide some citation for these assumptions.

4) section 3.3 authors say from simulation results that: “In the absence of treatment, disease prevalence is greater among young cats, and among them in males.” Is this result consistent with some field observation? Did the authors find evidence of this relationship in some reported study? If so, it can constitute an important piece of evidence supporting the model design.

5) Authors say that is important to treat cats when symptoms are appreciated. It would be important to mention in the manuscript which are those symptoms in cats, as well as in humans.

6) As the authors point out, this disease is strongly unreported in cats and humans, so the infecteds that come out from the model are not the “true” number of infecteds. How would be results modified if the under-report is taken into account?

Minor comments:

- Introduction, typo in first word Sporotrichosis.

- Please revise all the references, in particular 6,7,8,9,10 in which country names should start with capital letters. Also other refs where author names are all in capital letters like 10, 22, and 24 and should not be.

- Unify labels, numbers and size of the figures. For example make bigger letters and numbers of figs 3 and 4, to the size of fig. 2.

- section 3.2 end of paragraph assuming the population age distribution is (instead of “are”) at the steady…..

-in figs revise x axis labels, unify styles and units as (months).

-revise y axis fig 3 y 4 “fraction” instead of “number” of infected cats.

- insets with refs in graphs, revise capital letters unify criteria in all graphs.

6. PLOS authors have the option to publish the peer review history of their article (what does this mean?). If published, this will include your full peer review and any attached files.

Reviewer #1: No

Reviewer #2: No

---

## [Author Response · Author response to Decision Letter 0]

20 Nov 2022

Response to reviewers:

Reviewer #1: 

1) In page 4 the authors state that the probability of acquiring the infection is higher in females. Why?

In general, unneutered males are more affected by sporotrichosis due to behavioral habits. However, for this model, we assume that the probability of acquiring Sporothrix infection during mating is higher in females, because the bite of the dorsum of the neck, and the possibility of direct contact with exudate of cutaneous lesions in sick tom cats. Importantly, biting the skin of the neck is a remnant behavior used to immobilize the female and provide proper orientation for mounting.

We have added a shortened version of the above explanation, emphasizing that this probability is per contact. Males, however, have more risky contacts than females due to fighting behavior.

2) In section 3 the distribution of disease prevalence is obtained for the different groups considered in the model. Are there any empirical data on this for some comparison? 

In section 3.1, the age-structure in the demographic equilibrium is calculated from demographic parameters obtained from the literature (fertility rate, and environment's carrying capacity) and from interviews with specialists (mortality rates by age class). No a priori guess is made about disease prevalence. All prevalences reported in figures 3 and 4 as well as in table 2, are the results of simulations under different scenarios until the system reaches its endemic equilibrium.

3) In the pdf, just after the figure for the flow chart of the model, there is a figure without any caption, which is repeated a few pages later. Please provide a caption and refer it in the text properly.

This has been fixed. We apologize for any confusion it may have caused.

4) For me the main shortcoming of the paper is the absence of a discussion of how the control of disease is implemented in the model. Is it turned on at some moment in time and set at some constant rate? Is it implemented just once? How do they change the model equations? Section 3.4 should be expanded with some discussion and details on this. 

Multiple control scenarios are compared in the results (table 2) in terms of prevalence after 5 or 10 years of enforcement, Scenarios are combinations of pharmacological treatment of sick cats and Neutering at a pre-reproductive age. We have extended the description of the scenarios in section 3.4 to clarify the comparison of control scenarios.

Reviewer #2: Review

1) One of my main concern is regarding the mean field formulation of the model, because as the authors point out they took into account only domiciled cats (as opposed to stray cats), and therefore those cats might be highly clustered inside houses with very low interaction among them? Therefore the spatial distribution of cats might be highly relevant. Can the authors justify a bit more why the mean field approach is acceptable?

In this paper, we present a mathematical description of the dynamics of sporotrichosis transmission in a population of domestic cats, with street and neighborhood access, stratified by age and sex. The contact rates represent the average in the overall population. The mean field approach is justified because of the lack of the detailed characterization of the spatial clustering of cat populations. Moreover, the model also assumes that domiciled cats do have access to the street, which minimize the effects of clustering. 

We have extended the first paragraph justifying the usage of mean field models given domestic cat's mobility and social contact patterns. We have added an extra reference about these behavioral characteristics.

2) Related with the above concern. It would be of importance to describe at some point in the manuscript, the behavior and area of movement of these domiciled cats. Can they circulate outside their houses through the neighborhood? If so, they might encounter other domestic or stray cats on the way. Those stray cats are not considered into the model...

Otherwise, where do the encounters occur among domiciled cats? In the same house? How many cats are on average in houses? I think that it's necessary to include in the manuscript a brief answer to these questions because they are relevant to disease transmission.

As explained in our response to the previous comment, we do assume that domestic cats have almost unrestricted access to the street population, which is supported by the literature. Our study did not conduct a survey or census on any specific region, but we had conversations with colleagues that were experienced in such surveys that agreed with our assumptions.

3) page 4, in the model the probability of acquiring infection is assumed to be higher in females. Is there any reference the authors may cite to support this assumption? The same for the assumption that males have more sexual encounters than females. Please provide some citation for these assumptions. 

In general, unneutered males are more affected by sporotrichosis due to behavioral habits. However, for this model, we assume that the probability of acquiring Sporothrix infection during mating is higher in females, because the bite of the dorsum of the neck, and the possibility of direct contact with exudate of cutaneous lesions in sick tom cats. Importantly, biting the skin of the neck is a remnant behavior used to immobilize the female and provide proper orientation for mounting. In relation to the assumption that males have more sexual encounters than females, under controlled conditions, one tomcat is usually sufficient for 20 females. In addition, during mating season, territorial males become increasingly irritable and protective of their areas. This is partly because other males wander great distances, with less recognition of territories, interacting with several groups of females. The increased contact between males can result in increased intermale aggression, particularly during encounters between individuals sharing an area. Intermale aggression, which is controlled by testosterone, can be violent and even take precedence over sexual behavior.

https://veteriankey.com/male-feline-sexual-behavior/#bib11

4) section 3.3 authors say from simulation results that: "In the absence of treatment, disease prevalence is greater among young cats, and among them in males." Is this result consistent with some field observation? Did the authors find evidence of this relationship in some reported study? If so, it can constitute an important piece of evidence supporting the model design. 

 In studies on feline sporotrichosis conducted in Rio de Janeiro, most of non-treated cats were male and young adults (median age was 24 months) (Reis et al., 2012; Souza et al., 2018; Boechat et al., 2018; Miranda et al., 2018).

We have added these 4 references to the text to support this statement.

5) Authors say that is important to treat cats when symptoms are appreciated. It would be important to mention in the manuscript which are those symptoms in cats, as well as in humans. 

Feline sporotrichosis is clinically presented as a single lesion or as multiple cutaneous lesions (mainly nodules and ulcers). In addition, respiratory signs (eg.: sneezing, rhinorrhea and dyspnea) and lymph node enlargement are frequently observed. The most frequent clinical form is characterized by multiple cutaneous lesions with nasal mucosal involvement and respiratory signs (Gremião et al., 2021).

We have added this description to the Introduction for reference to interested readers.

---

## [Decision Letter · Decision Letter 1]

3 Jan 2023

Modelo Matemático da Dinâmica de Transmissão e Controle da Esporotricose em Gatos Domésticos

PONE-D-22-19278R1

Dear Dr. Araújo,

We’re pleased to inform you that your manuscript has been judged scientifically suitable for publication and will be formally accepted for publication once it meets all outstanding technical requirements.

Kind regards,

Sebastián Gonçalves, Ph.D.

Academic Editor

PLOS ONE

Additional Editor Comments (optional):

Reviewers' comments:

Reviewer's Responses to Questions

**Comments to the Author**

1. If the authors have adequately addressed your comments raised in a previous round of review and you feel that this manuscript is now acceptable for publication, you may indicate that here to bypass the “Comments to the Author” section, enter your conflict of interest statement in the “Confidential to Editor” section, and submit your "Accept" recommendation.

Reviewer #1: All comments have been addressed

Reviewer #2: All comments have been addressed

2. Is the manuscript technically sound, and do the data support the conclusions?

Reviewer #1: Yes

Reviewer #2: Yes

3. Has the statistical analysis been performed appropriately and rigorously? 

Reviewer #1: Yes

Reviewer #2: Yes

4. Have the authors made all data underlying the findings in their manuscript fully available?

Reviewer #1: Yes

Reviewer #2: Yes

5. Is the manuscript presented in an intelligible fashion and written in standard English?

Reviewer #1: Yes

Reviewer #2: Yes

6. Review Comments to the Author

Reviewer #1: I believe the new version of the paper is ready for acceptance. The questions raised were essentialy the same as the other reviewer. The new version now addresses the main issues pointed out.

Reviewer #2: The authors have addressed all my concerns raised in the previous round of review. I recommend the publication of the new version of the manuscript.

7. PLOS authors have the option to publish the peer review history of their article (what does this mean?). If published, this will include your full peer review and any attached files.

Reviewer #1: No

Reviewer #2: No

---

## [Editor Report · Acceptance letter]

25 Jan 2023

PONE-D-22-19278R1 

Mathematical model of the dynamics of transmission and control of Sporotrichosis in domestic cats  

Dear Dr. Araújo:

I'm pleased to inform you that your manuscript has been deemed suitable for publication in PLOS ONE. Congratulations! Your manuscript is now with our production department. 

Kind regards, 

on behalf of

Dr. Sebastián Gonçalves 

Academic Editor

PLOS ONE